# Assessment of Inorganic Phosphate Intake by the Measurement of the Phosphate/Urea Nitrogen Ratio in Urine

**DOI:** 10.3390/nu13020292

**Published:** 2021-01-20

**Authors:** María Victoria Pendón-Ruiz de Mier, Noemí Vergara, Cristian Rodelo-Haad, María Dolores López-Zamorano, Cristina Membrives-González, Rodrigo López-Baltanás, Juan Rafael Muñoz-Castañeda, Francisco Caravaca, Alejandro Martín-Malo, Arnold J. Felsenfeld, Eugenio J. De la Torre, Sagrario Soriano, Rafael Santamaría, Mariano Rodríguez

**Affiliations:** 1Maimonides Institute for Biomedical Research of Cordoba (IMIBIC), Reina Sofia University Hospital, Nephrology Service, University of Cordoba, 14004 Cordoba, Spain; mvictoriaprm@gmail.com (M.V.P.-R.d.M.); noemiverse@gmail.com (N.V.); crisroha@yahoo.com (C.R.-H.); cristina_cmg92@hotmail.com (C.M.-G.); Rodri_lb@hotmail.com (R.L.-B.); amartinma@senefro.org (A.M.-M.); marias.soriano.sspa@juntadeandalucia.es (S.S.); rsantamariao@gmail.com (R.S.); marianorodriguezportillo@gmail.com (M.R.); 2Spanish Renal Research Network (REDinREN), Institute of Health Carlos III, 28029 Madrid, Spain; 3Nephrology Service, Reina Sofia University Hospital, 14004 Cordoba, Spain; doloreslopezzamorano58@gmail.com; 4EUTOXandCKD-MBD groups of the ERA-EDTA, Spanish Renal Research Network (REDinREN), Institute of Health Carlos III, 28029 Madrid, Spain; 5Nephrology Service, Infanta Cristina Hospital, 06080 Badajoz, Spain; fcaravacam@senefro.org; 6Department of Medicine, Veterans Affairs Greater Los Angeles Healthcare System and the David Geffen School of Medicine, University of California, Los Angeles, CA 90073, USA; afelsenf@ucla.edu; 7General Medicine, Miguelturra Clinic, 13170 Miguelturra/Ciudad Real, Spain; eujo06@ono.com

**Keywords:** phosphaturia, phosphate intake, FGF23, PTH, CKD

## Abstract

In chronic kidney disease (CKD) patients, it would be desirable to reduce the intake of inorganic phosphate (P) rather than limit the intake of P contained in proteins. Urinary excretion of P should reflect intestinal absorption of P(inorganic plus protein-derived). The aim of the present study is to determine whether the ratio of urinary P to urinary urea nitrogen (P/UUN ratio) helps identify patients with a high intake of inorganic P.A cross-sectional study was performed in 71 patients affected by metabolic syndrome with CKD (stages 2–3) with normal serum P concentration. A 3-day dietary survey was performed to estimate the average daily amount and the source of P ingested. The daily intake of P was 1086.5 ± 361.3 mg/day; 64% contained in animal proteins, 22% in vegetable proteins, and 14% as inorganic P. The total amount of P ingested did not correlate with daily phosphaturia, but it did correlate with the P/UUN ratio (*p* < 0.018). Patients with the highest tertile of the P/UUN ratio >71.1 mg/g presented more abundant inorganic P intake (*p* < 0.038).The P/UUN ratio is suggested to be a marker of inorganic P intake. This finding might be useful in clinical practices to identify the source of dietary P and to make personalized dietary recommendations directed to reduce inorganic P intake.

## 1. Introduction

In the early stages of chronic kidney disease (CKD), an increase in fibroblast growth factor-23 (FGF23) and parathyroid hormone (PTH) cause a reduction in the tubular resorption of phosphate (P). Thus, the serum concentration of P is maintained within normal values despite the decrease in the glomerular filtration of P [1]. However, in advanced CKD (stages 4–5) the serum P concentration may be increased [2] because the phosphaturic effects of FGF23 and PTH are not sufficient to overcome the marked reduction in the glomerular filtration of P. Consequently, at some stage of CKD, a reasonable strategy would be to reduce the intestinal absorption of P in an attempt to prevent the development of hyperphosphatemia. Ideally, it would be desirable to reduce the intake of inorganic P, which is easily absorbed, rather than limiting the intake of P contained in proteins that are necessary to prevent poor nutrition and muscle wasting.

Almost all food contains P. A large amount of P is found in proteins of animal origin. Moreover, P salts (inorganic P) contained in some beverages and additives of processed foods are a major source of highly absorbable P [3,4,5,6]. It has been shown that the restriction of additives containing P contributes to the control of serum P [6]. The P contained in animal proteins is hydrolyzed and then absorbed; P from plant proteins is poorly absorbed since it is bound to phytate, and humans lack phytase, the enzyme required to extract P from phytate [7]. The intestinal absorption of P from animal protein ranges from 50% to 70%, and from vegetable proteins, the intestinal absorption is only 30% to 50%. In contrast, the absorption of inorganic P is close to 100% [3,8,9]. Hence, the net absorption of P [10] is influenced by several factors, such as total P intake, food source, and vitamin D levels, which increased the intestinal absorption of P [7,11]. Given all these factors affecting P absorption, it is not surprising to find reports showing that the urinary excretion of P does not correlate with total P intake [12]. Therefore, nephrologists may not have a strong reason to measure urinary excretion of P.

High serum values of P should be corrected because there is an association between high serum P and increased mortality both in CKD patients and in the general population [13,14,15,16]. Recently, we observed that in the early stages of CKD, P overload, as reflected by high urinary P excretion, induces renal injury and accelerates renal disease progression [17]. Other authors suggest that an increased P intake is associated with cardiovascular morbidity and mortality, even if the serum level of P remains within the normal range [18,19]. Therefore, even with normal serum P, it may be essential to reduce P load in CKD, because it would decrease the demand for PTH and FGF23. Recent reports indicate that an increased tubular load of P, required to maintain P balance, may contribute to the progression of CKD [5,17]. Furthermore, there is increasing evidence indicating that the elevation of FGF23, necessary to maintain serum P, is associated with negative cardiovascular outcome [20]. Considering that the restriction of dietary P is a desirable therapy in CKD patients, it would be important to have an easy method available to assess the source of dietary P and make personalized nutritional recommendations. Ideally, protein intake should be preserved and restrict the easily absorbable inorganic P.

Urinary excretion of P should reflect the intestinal absorption of P, which includes both inorganic and organic P (from proteins). It is well established that the content of urinary urea nitrogen (UUN) is a measure of the amount of protein that has been absorbed and metabolized [21]. Hypothetically, an excessive intake of inorganic P relative to the intake of organic P should be reflected in an increase in the ratio of P/UUN in the urine.

The aim of the present study is to determine whether the ratio of P to UUN in urine helps to differentiate the sources of P ingested in a homogeneous group of patients with metabolic syndrome and CKD stages 2–3.

## 2. Materials and Methods

### 2.1. Study Design

This is a cross-sectional study of patients being followed regularly at the outpatient kidney and hypertension clinic. All subjects gave their informed consent for inclusion in the study. The study was conducted in accordance with the Declaration of Helsinki, and the protocol was approved by the Ethics Committee of Cordoba (Cordoba Research Ethics Committee, Spain. Record number: 240; committee’sreferencenumber: 2730) and *Agencia Española de Medicamentos y Productos Sanitarios* (Spain. EudraCT number: 2015-000619-42).

### 2.2. Inclusion Criteria

The total number of patients included in the study was 71. There were 18 to 85 years old with an estimated glomerular filtration rate (eGFR) between 30 and 90 mL/min/1.73 m^2^ (CKD stages 2–3) [22]. All patients had been diagnosed with metabolic syndrome, fulfilling at least three of the following five criteria [23]: (1) waist circumference ≥88 cm in women and ≥102 cm in men; (2) serum triglycerides ≥1.69 mmol/L or drug treatment for hypertriglyceridemia; (3) low high-density lipoprotein; men <1.03 mmol/L, women <1.29 mmol/L or drug treatment; (4) elevated fasting glucose ( ≥6.11 mmol/L) or drug treatment for diabetes mellitus; and (5) systolic blood pressure ≥130 and/or diastolic blood pressure ≥85 mmHg or treatment for high blood pressure. All patients had limited proteinuria (albumin/creatinine ratio <0.4 mg/mg and a protein/creatinine ratio <1 mg/mg).

### 2.3. Exclusion Criteria

Patients with serum P concentration >1.61 mmol/L, congestive heart failure, glomerular disease, HIV infection, hepatitis B or C, chronic liver disease, systemic inflammatory disease, or previous history of cancer in the last 5 years were excluded.

### 2.4. Patients

Participant flow chart is shown in Figure 1. All patients were followed in the outpatient clinic by the same physician at least twice yearly. Baseline data included age, gender, body mass index, waist circumference, blood pressure, and comorbidities, or prevalent diseases. Hypertension was defined as a systolic blood pressure >140 mmHg and/or diastolic blood pressure >90 mmHg or a previous diagnosis of hypertension on blood pressure medication. Abnormalities in glucose metabolism were defined as a fasting glucose level >6.11 mmol/L or the use of hypoglycemic medication.

Patients underwent a detailed 3-day dietary survey prior to the extraction of blood and urine samples to estimate the average daily amount and the source of P ingested. The intake of P was estimated from a table with the composition of aliments consumed in Spain, particularly in the south of Spain [24]. This extensive table has been very helpful to obtain the P content of many homemade dishes that are common in this region. The information on the content of P shown in this table matched the values indicated in another source of food information, the Spanish Food Composition Database “BEDCA” [25] developed by the Spanish Federation of Food and Beverage Industries (“FIAB: *Federación Española de Industrias de la Alimentación y Bebidas*”) and the Spanish Agency for Food Safety and Nutrition (“AESAN”). The food composition values collected in this database have been obtained from different sources, including laboratories, food industry, and scientific publications. This database is built according to European standards developed by the EuroFIR European Network of Excellence, and is included in the list of food composition databases of the EuroFIR Association [26]. The dietary record has been used as an appropriate assessment method for a pan-European dietary survey of the European Food Safety Authority (EFSA). However, with the information available, it is not possible to estimate with precision the amount of inorganic P contained in the diet; the data collected and the values reported are only a brief estimation based in the amount of processed food ingested (self-reported) by the patients. Before obtaining dietary information, patients were trained on how to adequately describe the foods, amounts consumed, cooking methods, etc. Food consumption was recorded daily both per meal and between meals. Food quantities were assessed by a description of the utensils commonly used for cooking and serving and indicating the amount consumed of each product [27]. Patients were asked to report if, independently of the amount consumed, the diet included processed food. The survey was reviewed with the patient during the clinic visit with the aim of clarifying the entries and adding any omitted items and amounts.

### 2.5. Blood and Urine Chemistries

Blood was collected for measurements of serum biochemistry and complete blood count. Laboratory tests included serum creatinine, urea, glucose, uncorrected calcium, P, magnesium, albumin, lipids, ferritin, iron, and C reactive-protein (CRP). The 24-h urine samples were collected and used for quantification of P electrolytes, albumin, protein, urea with an Architect c-16000 (Abbott^®^, Chicago, IL, USA). The eGFR was calculated using the Chronic Kidney Disease Epidemiology Collaboration (CKD-EPI) formula [22]. Complete blood count was measured with a ABX Pentra 120 Retic^®^ (Horiba, Kyoto, Japan). Human plasma c-terminal FGF23 (c-FGF23) was determined by ELISA (Inmunotopics, San Clemente, CA, USA). Intact PTH level was quantified by ELISA (Inmunotopics, San Clemente, CA, USA).

Parameters measured in the 24-h urine collection were: Urine P excretion (mg/day) that should match daily intestinal absorption of P. Fractional excretion of P (FeP) (%) calculated as FeP = ([Serum creatinine × Urine P]/[Urine creatinine × Serum P]) × 100. Urine P excretion/eGFR (mg/day/eGFR), that reflects the urine P excretion relative to kidney function. Urine P/creatinine ratio (P/Cr ratio) (mg/mg). Urine Urea nitrogen (UUN) excretion (mg/day) (UUN (mg/dL) = Urea (mg/dL)/2.1428), UUN/Cr ratio (mg/mg). P/UUN ratio (mg/g) that reflects daily intestinal absorption of P relative to daily amount of absorbed and metabolized protein.

### 2.6. Animal Studies

To obtain additional information about the relationship between inorganic P intake and the ratio P/UUN in urine, we analyzed urine samples stored in our Biobank. These are samples from previous experiments using experimental rat model of 5/6 nephrectomy (Nx) fed diets with inorganic phosphate added to achieve a 0.2%, 0.4%, 0.6%, and 0.8% of total dietary content of phosphate. The normal P content in rat diet is 0.6%.

#### 2.6.1. Animals

Male Wistar rats (Charles River Laboratories, Wilmington, MA, USA), 9–10 weeks old and weighing 250–300 g, were individually housed using a 12-h/12-h light/dark cycle and given ad libitum access to 1318 Altromin breeding diet (Altromin, Germany). Rats received humane care in compliance with the Principles of Laboratory Animal Care formulated by the National Society for Medical Research. Ethics approval was obtained from the Ethics Committee for Animal Research of the University of Cordoba.

Renal failure was induced by subtotal nephrectomy (5/6 Nx), a two-step procedure. In the first step, animals were anesthetized using xylazine (5 mg/kg intraperitoneally) and ketamine (80 mg/kg intraperitoneally). A 5 to 8 mm incision was made on the left mediolateral surface of the abdomen. The left kidney was exposed, and the two poles (two thirds of the renal mass) were ablated. After 1 week, animals underwent right Nx under anesthesia. The control group underwent sham operation.

#### 2.6.2. Dietary Inorganic Phosphorus Modulation

One day after the second surgery, rats were randomly distributed into groups and fed diets with inorganic P added to achieve a 0.2%, 0.4%, 0.6%, and 0.8% of total dietary content of P for 15 days. During the last 3 days of the 15-day experiment, rats were housed in metabolic cages to collect urine samples. Urine Urea nitrogen and P were quantified by spectrophotometry (Biosystems, Barcelona, Spain).

### 2.7. Statistical Analysis

Continuous variables are shown as mean (± standard deviation, SD) or median (interquartile range, IQR). Categorical variables are presented as a percent (%). Simple correlation analysis (Spearman) was used to identify the relationship between phosphaturia variables and other variables. P/UUN ratio was categorized into tertiles to compare P intake, phosphaturia, FGF23, and PTH. The Kruskal–Wallis test was used to compare the difference between means of more than two groups. Multivariable logistic regression was used to evaluate the association between various factors and the odds of having consumed processed food as patients included were asked to answer whether they ingested known processed food in a binary fashion (Yes/No). Two models were performed, and model 2 was selected as the final model because it had the best goodness-of-fit. Discrimination ability was performed using the ROC (Receiver Operating Characteristic) curve and Youden’s Index identified the optimal cutpoint to discriminate patients whoself-reported having consumed processed food from those who did not. The area under the curve (AUC), sensitivity, specificity, and the accuracy of the model were also assessed. A *p*-value <0.05 was considered statistically significant. Statistical analyses were performed using SPSS statistical program (SPSS Inc., Chicago, IL, USA) and R 3.6.1 (R Core Team, 2019), the *tableOne* (v*0.10.0* [28]), the *corrplot* (V0.84 [29]), and *cutpoint packages* [30].

## 3. Results

Demographics and clinical characteristics of the 71 patients are shown in Table 1. Mean age 61 ± 9 years-old, (range 38 to 77), 72% (n = 51) were male, with a mean eGFR of 68 ± 28 mL/min/1.73 m^2^. Hypertension was present in 100% of patients (n = 71), insulin resistance in 49% (n = 35) and dyslipidemia in 77% (n = 55). The mean body mass index (BMI) was 32.5 ± 4.1 kg/m^2^, and the mean body weight was 90.6 ± 13.8 kg. Serum and urine biochemistry and calculated urine P parameters are shown in Table 2.

The average intake of P was 1086.5 ± 361.3 mg/day (range 286.2 to 2093.7 mg/day). The sources of P were: animal protein (64 ± 13%, 677.3 ± 215.1 mg/day), vegetable protein (22 ± 10%, 236.5 ± 130.2 mg/day), and inorganic P (14 ± 13%, 174.8 ± 205.1 mg/day).

Simple linear correlation analyses of daily intake of P versus urine parameters are shown in Table 3. There was no correlation between P intake and the 24-h Urine P excretion or any of the urinary P parameters, except with the P/UUN ratio in urine (*p* = 0.018). The P/UUN ratio reflects total P absorbed and excreted in urine relative to the amount of protein absorbed and metabolized. Linear correlation between P/UUN ratio in urine and different dietary sources of P is shown in the lower panel of Table 3. The urine P/UUN ratio showed a significant correlation with the amount of inorganic P ingested (*p* = 0.005) and failed to show a correlation with P intake from animal or vegetable origin.

These findings are not surprising since inorganic P augments the numerator of the P/UUN ratio, whereas animal or vegetable sources of P increase both the numerator and the denominator of the P/UUN ratio. Therefore, according to the results presented in Table 3, the P/UUN ratio, but not the daily phosphaturia, appear to reflect the intake of inorganic P.

To assess to what extent the different sources of P ingested account for the total P intake, the amount of P ingested from the different sources (animal, vegetable, and inorganic) was plotted against total P intake (Figure 2A–C, respectively). Patients within the highest tertile of P intake show a disproportionally high intake of inorganic P (Figure 2C) relative to the amount of P ingested of animal (Figure 2A) or vegetable (Figure 2B) origin. Therefore, patients within the highest tertile of P intake eat proportionally more inorganic P than P from proteins; thus, in these patients, it is expected to find an elevation in the P/UUN in the urine.

Subsequently, patients were categorized into tertiles of P/UUN ratio in urine to determine if patients with the higher P/UUN ratio show a high intake of inorganic P. Patients in the highest tertile of P/UUN ratio showed a significantly higher intake of P mainly from inorganic sources. Moreover, it exhibited more phosphaturia (Table 4). Although the average intake of inorganic P is substantially less than that of organic P, the intestinal absorption of inorganic P double that of organic P. The age and eGFR were similar in the different tertiles of P/UUN (Table 4).

The amount of inorganic P in the diet is not quantified with precision. However, we had available a self-reported questionnaire from patients indicating whether they had eaten processed foods, bakery, beverages, and others that are known to contain different amounts of inorganic P. Patients were asked to answer whether they ingested known processed food in a binary fashion (Yes/No). This is not an estimation of the amount of inorganic P ingested, it was just a yes/no answer. A logistic regression was performed to analyze if the urinary P/UUN ratio was associated with the probability of having consumed processed food. The mean (SD) values of P/UUN were: 71.4 ± 15.9 mg/g and 60.6 ± 14.9 mg/g, *p* =0.005, in patients who, respectively, reported having consumed processed food and those who did not. Results of logistic multivariable regression analysis using eat vs did not eat processed food as dependent variables showed the following information (Table 5):

P/UUN ratio is independently associated with the probability of having consumed P from inorganic sources. Age, on the contrary, is associated with the probability of having consumed P from other than inorganic P sources. The AUC of the model is 0.809 (Figure 3). Sensibility and specificity of the model were 82% and 62%, respectively. Youden’s index with bootstrap estimation for simulating the cutpoint variability (1000 replications) was used to estimate the optimal cutpoint to discriminate patients whoself-reported having consumed processed food from those who did not. Patients with a urine P/UUN ratio >67.3 mg/g have 80% probability of eating processed food.

The relationship between urine P parameters and phosphaturic hormones c-FGF23 and PTH was also analyzed. A simple linear correlation analysis is shown in Table 6. The 24-h urine P excretion, the urine P/Cr ratio, and the P/UUN ratio failed to show a significant correlation with c-FGF23 or PTH. However, eGFR, and the FeP correlated significantly with both c-FGF23 and PTH. Therefore, circulating levels of c-FGF23 and PTH correlated with the tubular load of P, but not with the total urinary excretion of P. The serum concentration of calcium, P, CRP, albumin, and iron did not show significant correlation with, c-FGF23 or PTH (data not shown).

To obtain additional information about the relationship between inorganic P intake and the urine ratio P/UUN ratio we used an experimental ratmodel of uremia by 5/6 Nxfed diets with increasing amounts of inorganic P (0.2%, 0.4%, 0.6% and0.8%). Figure 4 shows that urine P/UUN increases significantly with the percentage of inorganic P ingested. This result supports that the urine P/UUN ratio could be useful to evaluate an excessive intake of inorganic P.

## 4. Discussion

The present study shows that the daily intake of P does not correlate with the amount of P excreted in 24-h urine collection. However, it was found that the intake of inorganic P correlates with the ratio of P/UUN in urine and patients including processed food in their diets have an increase in P/UUN in urine. In our patients, an excessive intake of P is not entirely due to high protein intake; it is due to an excess of inorganic P intake. Results obtained from animal studies demonstrate that adding inorganic P to the diet produce a commensurate increase in the ratio P/UUN in urine. The study was performed in a uniform group of CKD stages 2–3 patients with metabolic syndrome and a normal serum P concentration. This study also analyzed whether serum levels of phosphaturic hormones (PTH and c-FGF23) were useful to evaluate phosphaturia. The serum concentrations of c-FGF23 and PTH did not correlate with total urinary P excretion but with tubular load of P, FeP, and eGFR. We learned from this study that the ratio of P/UUN could be a useful tool for clinicians to assess the intake of inorganic P and to make dietary recommendations.

Our results agree with previous findings by others [12], showing that phosphaturia measured in a 24-h urine collection does not correlate with the total amount of P ingested. Phosphaturia reflects the amount of P absorbed, which varies according to the source of P ingested. The intestinal absorption of P is lower from vegetables than from meat (30 to 50% vs. 50 to 70%) [10], whereas almost all inorganic P is absorbed. The proportion of P ingested from the different sources is not uniform among the different individuals; with such variability in factors affecting intestinal absorption of P, it is not a surprise to find no correlation between total P intake and the urinary excretion of P.

Protein malnutrition is frequent in CKD patients with advanced age. Protein restriction should not be recommended in patients with elevation of the P/UUN ratio. The total amount of inorganic P ingested is less than the P intake from animal and vegetable protein; however, it is almost totally absorbed and should contribute notably to the value of the numerator of the P/UUN ratio. The P intake from proteins of animal origin contributes to the numerator, but it also increases the amount of UUN, which tends to reduce the P/UUN ratio. It is widely accepted that the value of 24-h UUN excretion is a measure of the protein intake [31], so the P/UUN ratio reflects the amount of P excreted relative to the protein intake. It can be assumed that if protein intake is maintained and the amount of inorganic P ingested is decreased, the P/UUN ratio will be reduced. This hypothesis could be validated in future studies. From the results obtained in the present study, it is clear that if the patient has a high P/UUN ratio (>71.1 mg/g), the total P intake is high and with a high proportion of inorganic P (Figure 1, Table 4). Of interest is the study by Caravaca et al., showing that the reduction in intestinal absorption of P by P binders is reflected by a decrease in urine P/urea ratio [32].

Serum levels of PTH and c-FGF23 did not correlate with the total amount of P excreted. However, these phosphaturic hormones did correlate with the eGFR and the FeP, suggesting that it is the load of P relative to the prevailing GFR what dictates the stimulation of PTH and c-FGF23.

In our patients, it was not observed a correlation between P intake and phosphaturia. Others have reported the same observation. Twenty four-hour urine P excretion was considered a parameter that reflects P intake in a situation of P balance, assuming that net P absorption is linearly related to intake. This relationship has been assumed to apply to patients with CKD [33]. Ix et al. showed that the circadian pattern of serum P is modifiable by P intake [34], and the same author showed that P intake did not correlate with phosphaturia [12]. Stremke ER et al. have recently published that 24-h urine P was highly variable and was not correlated with dietary P intake in eight patients with moderate CKD on a tightly controlled dietary intake [35].

In the western diet, P is ingested primarily as protein. The recommended dietary allowance for P in our area is 900–1100 mg/day in healthy adults. The EFSA in Europe, reports that the P intake ranges should be between 1000–1767 mg/day [36]. Our patients have an average P intake of 1086.5 ± 361.3 mg/day, and only 14% is inorganic P, and much of this is contained in additives [7]. P ingested as additives are easily absorbed and excreted in the urine. The amount of P ingested as additives is low but contribute in large proportion to the P collected in urine obscuring the relationship with the total P intake. Despite their widespread use, P additives are typically unaccounted in the estimated P content of processed foods. P additives represent a significant and “hidden” P load in modern diets [3,37]. It must be recognized that the quantification of inorganic P in the diet is difficult because the foods do not have the necessary information to make the calculations. A high proportion of inorganic P is contained processed food. It was very interesting to observe that patients who reported the intake of processed food presented a high P/UUN in urine, in fact, if the value is >67.3 mg/g, they have 80% probability of having processed foods in their diets. We believe that this may be a valuable information in daily clinical practice. The results obtained from groups of uremic rats fed increasing amount of inorganic P indicate that maintaining P intake of animal or vegetable origin the addition of known amounts of inorganic P to the diet will produce predictable increases in the P/UUN ratio in urine. Of course, an increase in the P/UUN may be detected if the protein intake is reduced, which would be reflected by low amounts of urinary UUN excretion.

It is not clear whether dietary P should be restricted in CKD stages 2–3 normophosphatemic patients. Kidney Disease improving Global Outcomes (KDIGO) guidelines do not recommend the use of P binders in these patients if the serum P is not elevated. However, it is important to educate patients to select foods without P additives [38]. Certainly, a serum P within the high level of the normal laboratory range (<1.45 mmol/L) is a risk factor for vascular calcification [15]. An argument favoring the dietary restriction of specific sources of P is the fact that the tubular load of P is associated with an increase in both c-FGF23 and PTH, which is observed before there is a significant increase in serum P concentration. Reports indicate that a progressive increase in FGF23 predicts rapid progression of kidney disease and mortality [39].

The serum concentration of PTH and c-FGF23 correlated with the tubular load of P but did not correlate with the total amount of P in the urine. The degree of renal failure determines the amount FGF23 and PTH required to excrete an excess of P intake [11]. In animals, with renal failure, the increase in the renal load of P is accompanied by a reduction in tubular expression of klotho, which generates resistance to the phosphaturic effect of FGF23 [40]. The association between high levels of FGF23 and mortality [39] supports the importance of dietary restriction of inorganic P; therapeutic approaches to reduce the intestinal absorption of P, according to urinary P/UUN ratio, may prevent the increase in FGF23 [7,41]. Some authors have shown that short-term dietary P restriction tends to reduce FGF23 levels in patients with moderately decreased kidney function [42].

This study has several limitations. A high P load may be associated with a decrease in serum bicarbonate level, unfortunately serum bicarbonate was not measured. Moreover, it was not taken into account that patients on proton-pump inhibitors may have changed the intestinal absorption of P [43,44]. An important limitation of this type of study is the difficulty of a precise calculation of the P content in the diet despite a conscious training of patients in how to complete the dietary survey. Databases of food composition do not specify with enough precision the amount of P contained in food additives, a main source of inorganic P.

It was decided to performed this study in patients with CKD stages 2–3 with the same cause of renal failure in an attempt to collect more uniform results; however, a reasonable question is whether the results could be unrestrictedly applied to the general population. Nonetheless it is likely that the data obtained in these patients with metabolic syndrome could be useful in the general population.

## 5. Conclusions

In conclusion, our results suggest that the proportion between P and UUN excreted in urine increases with the relative amount of inorganic P ingested. Thus, the P/UUN ratio in 24-h urine is a marker of inorganic P intake, which is easily absorbed. It remains to be proven if, in daily clinical practice, the use of P/UUN ratio urine is useful to guide the dietary advice of patients with CKD.

## Figures and Tables

**Figure 1 nutrients-13-00292-f001:**
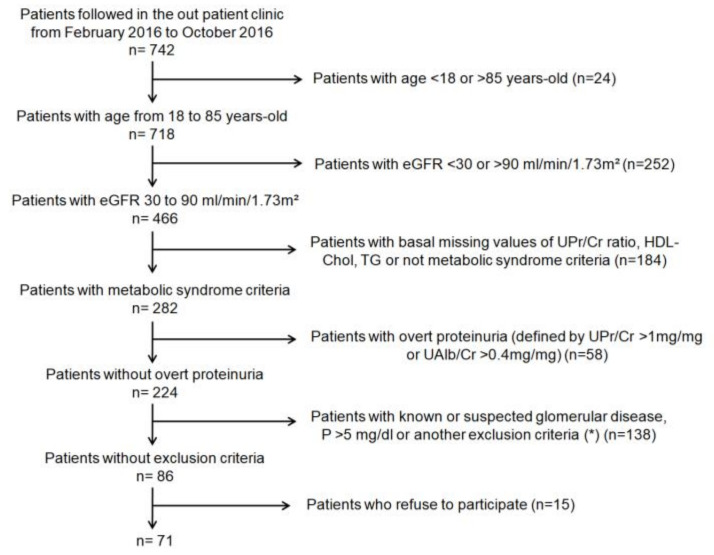
Participant flow chart.eGFR: estimated glomerular filtration rate; UPr/Cr ratio: ratio of protein to creatinine in urine; HDL-Chol: high-density lipoprotein cholesterol; TG: triglycerides; UAlb/Cr: ratio of albumin to creatinine in urine; P: phosphate. (*****) Congestive heart failure, HIV infection, hepatitis B or C, chronic liver disease, systemic inflammatory disease or previous history of cancer in the last 5 years.

**Figure 2 nutrients-13-00292-f002:**
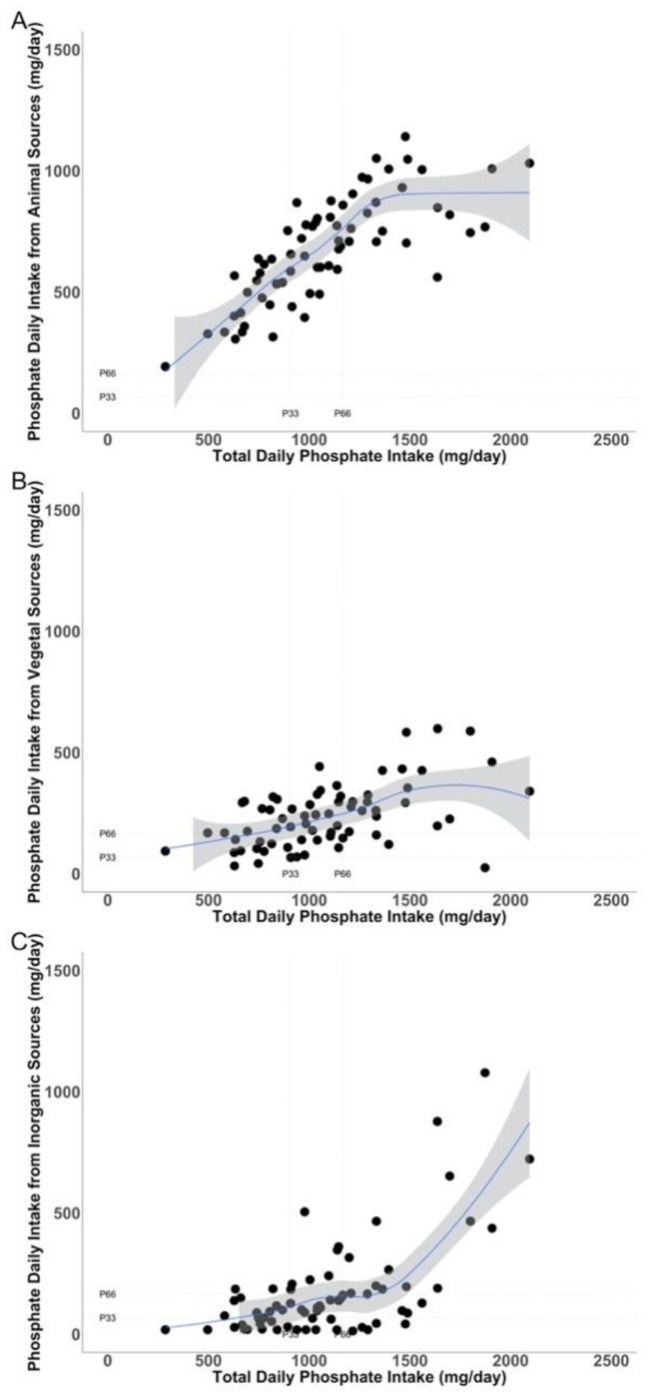
Relationship between the amounts of phosphate intake (mg/day) from different sources (*Y* axis):animal (**A**), vegetal (**B**), and inorganic (**C**), and the total phosphate intake (mg/day) (*X* axis).Tertiles of phosphate intake and the tertiles of the different sources of phosphate are shown as dotted line in the *Y* and *X* axis. P: phosphate.

**Figure 3 nutrients-13-00292-f003:**
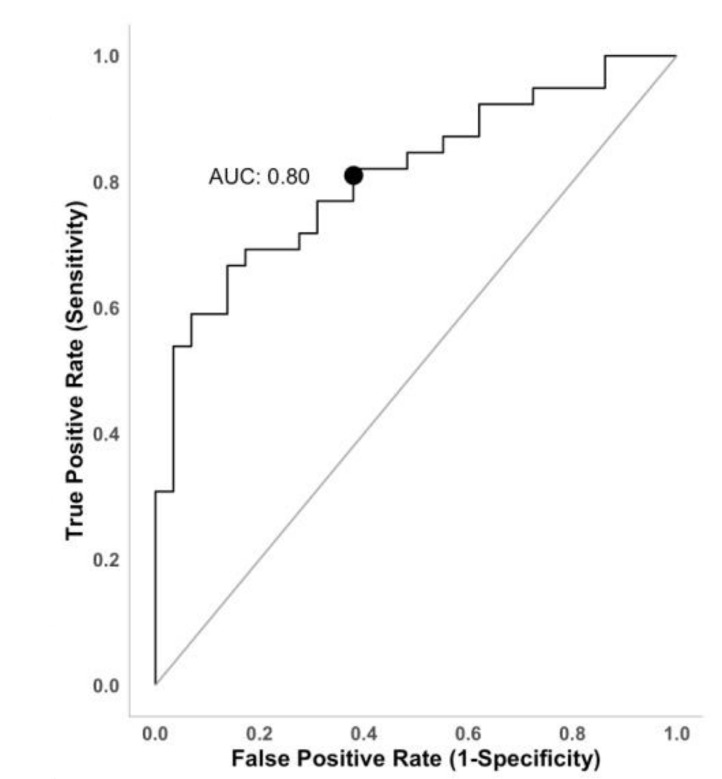
ROC curve representing the diagnostic accuracy of model 2.Area under the curve (AUC) = 0.80. The sensitivity of the model was 82% meaning the ability of the model to properly recognize the subjects that consumed processed food. Specificity was 62%, indicating that 62 out of 100 non-processed food consumers were properly identified. The accuracy of the model was 75%.

**Figure 4 nutrients-13-00292-f004:**
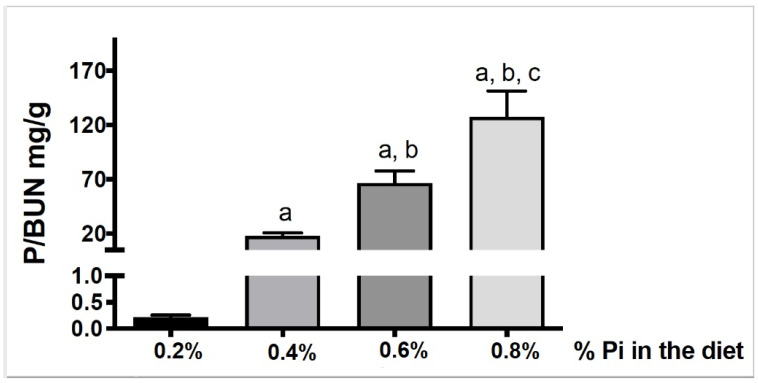
Relationship between inorganic P intake and P/UUN ratio. P/UUN was significantly higher as inorganic phosphate intake increases. a *p* < 0.001 vs. 0.2% group. b *p* < 0.001 vs. 0.4% group. c *p* < 0.01 vs. 0.6% group. Values are mean ± standard deviation.

**Table 1 nutrients-13-00292-t001:** Demographics and clinical characteristics of the study patients (n = 71).

Characteristic	Mean ± SD
Age (years)	61 ± 9
Gender (male, n/%)	51/72
Type 2 diabetes (n/%)	35/49
Dyslipidemia (n/%)	55.0/77
Hyperuricemia (n/%)	43.0/61
Smokers (n/%)	8.0/11
Ischemic cardiopathy (n/%)	8.0/11
Peripheral arteriopathy (n/%)	9.0/13
Cardiovascular events (n/%)	12.0/17
Hypertension (n/%)	71.0/100
ACEI (n/%)	7.0/10
ARB (n/%)	61.0/86
Diuretics (n/%)	41.0/58
CCB (D) (n/%)	45.0/63
Beta-blocker (n/%)	28.0/39
Central Alfa-Blocker (n/%)	31.0 /44
Other antihypertensive drugs (n/%)	4.0/6
Body weight (kg)	90.6 ± 13.0
Body mass index (kg/m^2^)	32.0 ± 4.0
Waist circumference (cm) in men	110.3 ± 10.1
Waist circumference (cm) in women	108.0 ± 10.1
Systolic blood pressure (mmHg)	136.3 ± 25.5
Diastolic blood pressure (mmHg)	84.1 ± 16.6
Mean blood pressure (mmHg)	108.4 ± 19.0
Heart rate (bpm)	64.0 ± 13.3

Quantitative and qualitative variables are presented as mean ± standard deviation and number/percentage, respectively. ACEI: angiotensin converting enzyme inhibitor; ARB: angiotensin receptor blocker; CCB (D): calcium channels blocker dihydropyridine.

**Table 2 nutrients-13-00292-t002:** Biochemical parameters for serum and 24-h urine.

Serum	Mean ± SD
Creatinine (mmol/L)	0.105 ± 0.036
Urea (mmol/L)	17.1 ± 5.7
Glucose (mmol/L)	7.14 ± 2.71
Sodium (mEq/L)	139.4 ± 2.4
Potassium (mEq/L)	4.4 ± 0.5
Chloride (mEq/L)	106.1 ± 3.3
Calcium (mmol/L)	2.37 ± 0.07
Phosphate (mmol/L)	1 ± 0.16
Magnesium (mmol/L)	0.8 ± 0.1
Triglycerides (mmol/L)	1.70 ± 0.74
Cholesterol (mmol/L)	4.60 ± 0.96
HDL-Cholesterol (mmol/L)	1.01 ± 0.23
LDL-Cholesterol (mmol/L)	2.80 ± 0.83
Albumin (µmol/L)	652 ± 30
CRP (µg/L)	26,000 (12,000–55,000)
Iron (µmol/L)	14.9 ± 6
Ferritin (µg/L)	74 (39–130)
PTH (pmol/L)	6.1 (4.2–8.5)
c-FGF23 (RU/mL)	77.0 (61.8–112.6)
**URINE (24-h Collection)**	**Mean ± SD**
Creatinine (mg/day)	1402.3 ± 618.4
UUN (g/day)	12.2 ± 4.5
Sodium (mEq/day)	183 ± 76
Potassium (mEq/day)	67 ± 28
Calcium (mg/day)	106 ± 79
**Urine P Parameters (24-h Collection)**	**Mean ± SD**
Urine Phosphate excretion (mg/day)	797.2 ± 324.7
Fractional excretion of phosphate (FeP) (%)	23.3 ± 10.0
Urine Phosphate excretion/eGFR (mg/day/eGFR)	13.1 ± 7.1
Phosphate/Creatinine ratio (mg/mg)	0.60 ± 0.17
Phosphate/UUN ratio (mg/g)	66.5 ± 16.3

Biochemical parameters for serum and 24-h urine summarized as mean ± standard deviation. c-FGF23: c-terminal fibroblast growth factor-23; CRP: C-reactive-protein; HDL: high density lipoprotein; LDL: low density lipoprotein; P: phosphate; PTH: parathyroid hormone; UUN: urine urea nitrogen.

**Table 3 nutrients-13-00292-t003:** Simple linear correlation analysis.

Variable	Spearman Coefficient	Significance (*p*)
(A) Total P intake (mg/day)		
Urine P Excretion (mg/day)	0.189	0.128
FeP (%)	−0.088	0.484
Urine P Excretion/eGFR (mg/day/GFR)	−0.032	0.797
P/Creatinine ratio (mg/mg)	0.110	0.373
UUN (g/day)	−0.11	0.927
UUN/Creatinine ratio(mg/mg)	−0.088	0.471
P/UUN ratio (mg/g)	0.284	0.018
(B) P/UUN ratio (mg/g)		
P from Animal protein (mg)	0.160	0.189
P from Vegetable protein (mg)	0.223	0.066
Inorganic P (mg)	0.332	0.005

(A) Total phosphate intake (mg/day) (dependent variable) vs.independent variables, measured in 24-h urine collection. (B) Phosphate/UUN ratio (mg/mg/g) (dependent variable) vs.independent variables: phosphate ingested from animal protein; phosphate ingested from vegetable protein and inorganic phosphate ingested. P: phosphate; eGFR: estimated glomerular filtration rate using CKD-EPI; FeP: fractional excretion of phosphate; UUN: urine urea nitrogen.

**Table 4 nutrients-13-00292-t004:** Patients separated by tertiles of urine phosphate/urine urea nitrogen ratio. Comparisons of age; glomerular filtration; total intake of phosphate; phosphate intake from animal, vegetable and inorganic sources; fractional excretion of phosphate; urine phosphate excretion per day.

Variable	T1 (<58.9 mg/g) n = 24	T2 (58.9–71.1 mg/g) n = 23	T3 (>71.1 mg/g) n = 24	*p*
Age (years)	62 ± 10	62 ± 8	61 ± 9	0.863
GFR (CKD-EPI) (mL/min/1.73 m^2^)	65 ± 23	66 ± 20	73 ± 19	0.441
Total intake of P (mg/day)	979.3 ± 348.5	1042.1 ± 355.2	1227.2 ± 347.3	0.047
P intake from animal source (mg/day)	628.5 ± 98.9	673.0 ± 726.2	726.2 ± 164.8	0.243
P intake from vegetable source (mg/day)	194.0 ± 98.9	257.3 ± 143.9	255.4 ± 137.2	0.185
Inorganic P intake (mg/day)	160.1 ± 211.1	113.4 ± 103.6	247.1 ± 253.1	0.038
FeP (%)	22.9 ± 12.3	24.1 ± 10.6	22.9 ± 6.8	0.653
Urine P Excretion (mg/day)	696.7 ± 237.2	770.1 ± 313.7	923.6 ± 356.1	0.049

FeP: fractional excretion of phosphate; GFR: glomerular filtration; P: phosphate; P/UUN ratio: phosphate/urine urea nitrogen ratio.

**Table 5 nutrients-13-00292-t005:** Multivariable logistic regression showing the factors associated with the probability of having consumed processed food.

		Model 1			Model 2	
Characteristic	OR	95 % CI	*p*-Value	OR	95% CI	*p*-Value
Gender (F vs. M)	0.939	0.232, 3.696	0.93			
Age (Years)	0.885	0.811, 0.95	**0.002**	0.883	0.807, 0.95	**0.003**
P/UUN ratio (mg/g)	1.058	1.015, 1.111	**0.013**	1.061	1.019, 1.112	**0.007**
P from animal sources (mg/day)	1.000	0.997, 1.003	0.82			
P from vegetal sources (mg/day)	1.002	0.997, 1007	0.52			
GFR (CKD-EPI) (mL/min/1.73 m^2^)				1.000	0.97, 1.029	0.99

Model 1: adjusted for age, gender, P/UUN ratio, daily P intake from animal and vegetable sources. Model 2: adjusted for ageP/UUN ratio, and eGFR by CKD-EPI equation. The bold means *p* < 0.05. OR: odds ratio; CI: confidence interval; F: female; M: male.

**Table 6 nutrients-13-00292-t006:** Simple linear correlation analysis between phosphate parameters obtained from 24-h urine collection and phosphaturic hormones.

Variable	Spearman Coefficient	Significance (*p*)
**Urine P excretion (mg/day)**		
c-FGF23 (RU/mL)	−0.149	0.225
PTH (pg/mL)	−0.096	0.435
**Urine P/Cr ratio (mg/mg)**		
c-FGF23 (RU/mL)	−0.009	0.938
PTH (pg/mL)	0.148	0.221
**Urine P/UUN ratio (mg/g)**		
c-FGF23 (RU/mL)	−0.022	0.859
PTH (pg/mL)	0.147	0.228
**eGFR (CKD-EPI)**		
c-FGF23 (RU/mL)	−0.534	<0.001
PTH (pg/mL)	−0.265	0.050
**FeP (%)**		
c-FGF23 (RU/mL)	0.367	0.002
PTH (pg/mL)	0.381	0.001

Fibroblast growth factor 23 (c-FGF23) and parathyroid hormone (PTH). Dependent variables: urine P excretion (mg/day): urine phosphate excretion (mg/day). Urine P/Cr ratio (mg/mg): urine phosphate/creatinine ratio (mg/mg)). Urine P/UUN ratio (mg/g): urine phosphate/urine urea nitrogen ratio (mg/mg). eGFR (CKD-EPI): glomerular filtration rate estimated by CKD-EPI (mL/min/1.73 m^2^). FeP (%): fractional excretion of P (%).

## Data Availability

Data is available on request.

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
