# Peer review of "Assessment of Inorganic Phosphate Intake by the Measurement of the Phosphate/Urea Nitrogen Ratio in Urine"

_nutrients, 2021, doi:10.3390/nu13020292_

Round 1

Reviewer 1 Report

The submitted paper is an exceedingly well written and genuinely relevant paper for a practicing nephrologist. Overall English is outstanding.  While the results are not unexpected it is nice to see our pre-existing strong clinical suspicion so well confirmed – phosphaturia is expected to correlate (or, rather, equals with) effective absorption. In essence, the study demonstrated the key importance of absorbable, rather than theoretically available P to define urine P appearance. In the manuscript, the Authors examined, in great details, correlations between dietary intake registered P intake and serum/urine P values, as well correlations with numerous P-regulating hormones in a modest-sized stage 2-3 CKD + complicated obesity (metabolic sy.) subjects. The cohort description is very meticulous and detail-oriented

The largest weakness of this paper is perhaps the relatively weakly written Abstract, missing the opportunity to effectively “advertise” the paper.

Abstract. Missing some key finding of the project prominently featured; currently, more descriptive and listing only some of the cohort’s baseline characteristics. Would add some key results/correlation – e.g., to show key finding of Table 4. /highest tertile correlation with urinary P appearance; key p values of critical association s to drive hiem the Authors main massages, etc.

It may hep – both for the main paper’ Results and the Abstract of Authors could recover 1-2 key values and parameters, which could “help” practicing clinicians and nutritionist to identify likely excess intake of inorganic phosphorus. Moreover, this reviewer cannot help wondering, whether the high intake subjects had a lower serum bicarbonate (HCO3) – so many electrolytes are reported but not the serum HCO3 (Table 2)

Discussion. One concern, perhaps insufficiently entertained by the Authors, is the potential of proton-pump inhibitors to interfere calcium (and lanthanum) carbonate exposure to reduced P-absorption (see: Ren Fail. 2020 Nov;42(1):799-806 [PMID: 32779954] and Am J Med Sci. 2017 Jan;353(1):82-8 [PMID: 28104108].  Presence of GERD/medical therapy was apparently not an exclusion criteria… and should be listed among limitations.

Bibliography. Correct to English, the references (still in Spanish referencing format version)

Author Response

Editors-in-Chief

Nutrients

December 2020

Dear Editors-in-Chief,

The manuscript "Assessment of inorganic phosphate intake by the measurement of the phosphate/urea nitrogen ratio in urine" (nutrients-1023894) has been reviewed and a number of comments and questions have been formulated; all of them are very reasonable and will certainly help to improve the manuscript.

Reviewer reports:

* Reviewer 1:

Comments and Suggestions for Authors

The submitted paper is an exceedingly well written and genuinely relevant paper for a practicing nephrologist. Overall English is outstanding. While the results are not unexpected it is nice to see our pre-existing strong clinical suspicion so well confirmed – phosphaturia is expected to correlate (or, rather, equals with) effective absorption. In essence, the study demonstrated the key importance of absorbable, rather than theoretically available P to define urine P appearance. In the manuscript, the Authors examined, in great details, correlations between dietary intake registered P intake and serum/urine P values, as well correlations with numerous P-regulating hormones in a modest-sized stage 2-3 CKD + complicated obesity (metabolic sy.) subjects. The cohort description is very meticulous and detail-oriented

The largest weakness of this paper is perhaps the relatively weakly written Abstract, missing the opportunity to effectively “advertise” the paper.

Abstract. Missing some key finding of the project prominently featured; currently, more descriptive and listing only some of the cohort’s baseline characteristics. Would add some key results/correlation – e.g., to show key finding of Table 4. /highest tertile correlation with urinary P appearance; key p values of critical association s to drive hiem the Authors main massages, etc.

It may hep – both for the main paper’ Results and the Abstract of Authors could recover 1-2 key values and parameters, which could “help” practicing clinicians and nutritionist to identify likely excess intake of inorganic phosphorus.

The authors appreciate the suggestions made by the reviewer. We agree with the reviewer, the abstract could be improved and therefore it has been modified according to the suggestions by the reviewer. Key results have been added to the abstract to make it more informative and appealing. Limitation of words allowed for the abstract prevented us from including more information.

Moreover, this reviewer cannot help wondering, whether the high intake subjects had a lower serum bicarbonate (HCO3) – so many electrolytes are reported but not the serum HCO3 (Table 2).

Yes, this is a very relevant question. A high phosphate load may be associated with a decrease in serum bicarbonate level, unfortunately serum bicarbonate was not measured. We think that a sentence should be added to the discussion to acknowledge that serum bicarbonate levels were not measured. (Lines 384-386)

Discussion. One concern, perhaps insufficiently entertained by the Authors, is the potential of proton-pump inhibitors to interfere calcium (and lanthanum) carbonate exposure to reduced P-absorption (see: Ren Fail. 2020 Nov;42(1):799-806 [PMID: 32779954] and Am J Med Sci. 2017 Jan;353(1):82-8 [PMID: 28104108]. Presence of GERD/medical therapy was apparently not an exclusion criteria… and should be listed among limitations.

We agree with the reviewer about the potential role of proton-pump inhibitors. This is an important variable that was not taken into account in the original manuscript. A limitation of the study is that the presence of gastroesophageal reflux or medical therapy with proton-pump inhibitors were not included as exclusion criteria. The following information has been included in the new version of the manuscript: “Also, it was not taken into account that patients on proton-pump inhibitors may have changed the intestinal absorption of phosphate” (Lines 386-387).

Bibliography. Correct to English, the references (still in Spanish referencing format version)

The reference 24 has been written in English.

Reviewer 2 Report

The investigators have submitted a very carefully done and well considered manuscript testing the hypothesis that the measurement of urine P/UUN would identify individuals ingesting a high inorganic phosphate diet. 

Strengths

The rationale and background for embarking on this study are very well laid out.  Specifically the pitfalls with using the usual methodology for assessing electrolyte balance, i.e., fractional excretion or 24h urine measurement, to determine phosphate intake are clearly delineated and the authors use this to justify the study well.

The methods are clearly laid out.  Training the subjects on description of their food intake prior to initiating the dietary recall data is an important step.

The inclusion and exclusion criteria are clear.

The correlation of urine Pi with multiple parameters of phosphate homeostasis is also informative.

For the most part, the conclusions are supported by the data.

Areas to improve

This is a very carefully performed study; however, as with any study, especially human-based, addressing potential limitations of the study is an important exercise.  For this study, the subjects all had metabolic syndrome.  Do the authors think that the results from this population would apply to the general population? 

The authors did attempt to standardize and accurately assess dietary phosphate intake; however, even the best efforts could fall short and this should be acknowledged by the authors. Having subjects on a defined metabolic diet would be ideal.

it is not clear why the authors expressed the relationship between urine Pi/UUN in tertiles and not continuously.  They comment on the positive correlation between deduced inorganic Pi intake at the highest tertile.  What about the lower levels?

How do the authors explain the fact that 24h urine Pi does not correlate with FGF23 but fractional excretion of Pi does?  What does this say about the relationship between dietary Pi, intestinal Pi uptake, renal Pi excretion, and FGF23?

Why is there no correlation with iPTH?

The authors include considerable demographic and laboratory data that was not addressed.  Were there any correlations between the "degree of metabolic syndrome" and the estimated inorganic Pi absorption?

Minor issues

1.  In the abstract, suggest change the word proteic to protein-derived.

2.  Under methods, line 140, suggest change the word extraction to collection

Author Response

Editors-in-Chief

Nutrients

December 2020

Dear Editors-in-Chief,

The manuscript "Assessment of inorganic phosphate intake by the measurement of the phosphate/urea nitrogen ratio in urine" (nutrients-1023894) has been reviewed and a number of comments and questions have been formulated; all of them are very reasonable and will certainly help to improve the manuscript.

Reviewer reports:

* Reviewer 2:

Comments and Suggestions for Authors

The investigators have submitted a very carefully done and well considered manuscript testing the hypothesis that the measurement of urine P/UUN would identify individuals ingesting a high inorganic phosphate diet.

Strengths The rationale and background for embarking on this study are very well laid out. Specifically the pitfalls with using the usual methodology for assessing electrolyte balance, i.e., fractional excretion or 24h urine measurement, to determine phosphate intake are clearly delineated and the authors use this to justify the study well.

The methods are clearly laid out. Training the subjects on description of their food intake prior to initiating the dietary recall data is an important step.

The inclusion and exclusion criteria are clear.

The correlation of urine Pi with multiple parameters of phosphate homeostasis is also informative.

For the most part, the conclusions are supported by the data.

The authors are thankful for these positive comments

Areas to improve: This is a very carefully performed study; however, as with any study, especially human-based, addressing potential limitations of the study is an important exercise. For this study, the subjects all had metabolic syndrome. Do the authors think that the results from this population would apply to the general population?

As suggested by the reviewer, a potential limitation of the present study could be the characteristics of the studied cohort: patients were followed as outpatients and all had metabolic syndrome. This provides uniformity of the patient sample. The following paragraph has been added to the new version of the manuscript: It was decided to performed this study in patients with CKD stages 2-3 with the same cause of renal failure in an attempt to collect more uniform results; however, a reasonable question is whether the results could be unrestrictedly applied to the general population. Nonetheless it is likely that the data obtained in these patients with metabolic syndrome could be useful in the general population.” (Lines 393-397).

The authors did attempt to standardize and accurately assess dietary phosphate intake; however, even the best efforts could fall short and this should be acknowledged by the authors. Having subjects on a defined metabolic diet would be ideal

We agree with the reviewer. Another limitation of this type of study is the difficulty in estimating phosphate intake. Therefore, emphasis was placed on the correct completion of the dietary survey and on the subsequent analysis of the estimation of phosphate intake. In fact, the method used to quantify phosphate intake was validated using an additional data base of food composition. This information has been included in the new version of the manuscript: An important limitation of this type of study is the difficulty of a precise calculation of the phosphate content in the diet despite a conscious training of patients in how to complete the dietary survey. Databases of food composition do not specify with enough precision the amount of phosphate contained in food additives, a main source of inorganic phosphate.” (Lines 387-392).

Given the importance of estimation of the phosphate in the diet we have decided to include additional information to the methods, so the procedure is explained with sufficient detail (Lines 169-182).

Certainly it would be ideal to have subjects on a defined metabolic diet. As a matter of fact a next protocol will be maintaining the diet during a 3 day period (control) and then give an oral supplement of phosphate for 3 additional days to analyze the change in P/UUN in urine.

It is not clear why the authors expressed the relationship between urine Pi/UUN in tertiles and not continuously. They comment on the positive correlation between deduced inorganic Pi intake at the highest tertile. What about the lower levels?

There was an association between increased inorganic phosphate intake and the highest tertile of P/UUN. But, there was no significant difference in inorganic phosphate intake between tertile 1 and 2 of P/UUN. Nevertheless, simple linear correlation between inorganic phosphate intake and P/UUN is statistically significant (Table 3).

How do the authors explain the fact that 24h urine Pi does not correlate with FGF23 but fractional excretion of Pi does? What does this say about the relationship between dietary Pi, intestinal Pi uptake, renal Pi excretion, and FGF23?

The serum levels of FGF23 are increased in patients with reduced GFR. In these patients the filtration of phosphate is decreased, the elevation of FGF23 inhibits proximal reabsorption of phosphate to induce phosphaturia, so the phosphate balance is maintained. Thus FGF23 correlates with the amount of phosphate excreted in urine relative to the amount of filtered phosphate (FeP).

Why is there no correlation with iPTH?

PTH reduces the tubular reabsorption of phosphate. PTH increase in CKD and as mentioned with FGF23; it correlates with the amount of phosphaturia relative to the amount of filtered P (FeP).

The authors include considerable demographic and laboratory data that was not addressed. Were there any correlations between the "degree of metabolic syndrome" and the estimated inorganic Pi absorption?

The group of patients included in this study fulfills the criteria of Metabolic Syndrome and they were affected to a similar degree. Thus it was not possible to find an association between the severity of Metabolic Syndrome and the estimated inorganic phosphate absorption.

Minor issues

1. In the abstract, suggest change the word proteic to protein-derived.

2. Under methods, line 140, suggest change the word extraction to collection

Thanks for these appreciations. The corrections have been done.

Reviewer 3 Report

This work deals with an area of great clinical significance and with many yet unsolved questions that the authors develop very well in the introduction. As the authors state, 24 hour urine phosphorus (P) reflects net GI P absorption which is very difficult to determine clinically in view of the different bioavailability of P in different foods. In a very reasonable way, the authors hypothesized that an excessive intake of inorganic P relative to the intake of organic P should be reflected by an increase in the ratio of P to urea nitrogen in the urine (P/UUN ratio). Therefore, they designed a study to determine whether the P/UUN ratio helps to differentiate the sources of P ingested in a homogeneous group of patients with metabolic syndrome and CKD stages 2-3.

Main concern:

How did they determine the inorganic P content from 3-day recalls? I can understand differentiation of animal and plant based phosphorus, but what database has the “inorganic” content of P available? The authors cite reference 24, which unfortunately I have no access to (Mataix Verdú, J. Tabla de composición de alimentos. 5th Ed. Granada, Spain: University of Granada; 2009). I doubt, however, that there exists such a database with the specific content of inorganic P in a good number of food items. This has been an ongoing issue in this field and a major problem is precisely how to estimate the inorganic phosphorus content of food.

This is my big issue with this study; how did they estimate the inorganic P content in food? All their further conclusions are based on that information being valid. If they prove they have that information, I believe this would be an important addition to the literature, Until then, I am not sure, this work adds anything concrete except some discussion on the subject.

Author Response

Editors-in-Chief

Nutrients

December 2020

Dear Editors-in-Chief,

The manuscript "Assessment of inorganic phosphate intake by the measurement of the phosphate/urea nitrogen ratio in urine" (nutrients-1023894) has been reviewed and a number of comments and questions have been formulated; all of them are very reasonable and will certainly help to improve the manuscript.

Reviewer reports:

* Reviewer 3:

Comments and Suggestions for Authors

This work deals with an area of great clinical significance and with many yet unsolved questions that the authors develop very well in the introduction. As the authors state, 24 hour urine phosphorus (P) reflects net GI P absorption which is very difficult to determine clinically in view of the different bioavailability of P in different foods. In a very reasonable way, the authors hypothesized that an excessive intake of inorganic P relative to the intake of organic P should be reflected by an increase in the ratio of P to urea nitrogen in the urine (P/UUN ratio). Therefore, they designed a study to determine whether the P/UUN ratio helps to differentiate the sources of P ingested in a homogeneous group of patients with metabolic syndrome and CKD stages 2-3.

Main concern:

How did they determine the inorganic P content from 3-day recalls? I can understand differentiation of animal and plant based phosphorus, but what database has the “inorganic” content of P available? The authors cite reference 24, which unfortunately I have no access to (Mataix Verdú, J. Tabla de composición de alimentos. 5th Ed. Granada, Spain: University of Granada; 2009). I doubt, however, that there exists such a database with the specific content of inorganic P in a good number of food items. This has been an ongoing issue in this field and a major problem is precisely how to estimate the inorganic phosphorus content of food.

This is my big issue with this study; how did they estimate the inorganic P content in food? All their further conclusions are based on that information being valid. If they prove they have that information, I believe this would be an important addition to the literature, Until then, I am not sure, this work adds anything concrete except some discussion on the subject.

We completely agree with the reviewer; there is not a data base available indicating the exact content of inorganic phosphate in different foods.

The reference 24 is a table with the composition of aliments consumed in Spain, particularly in the south of Spain. This extensive table has been very helpful to obtain the phosphate content of many homemade dishes that are common in this region. “The information on the content of phosphate shown in this table matched the values indicated in another source of food information, the Spanish Food Composition Database (“BEDCA”) (www.bedca.net/bdpub/) database developed by the Spanish Federation of Food and Beverage Industries (“FIAB: Federación Española de Industrias de la Alimentación y Bebidas”) and the Spanish Agency for Food Safety and Nutrition (“AESAN”) The food composition values collected in this database have been obtained from different sources including laboratories, food industry and scientific publications. This database is built according to European standards developed by the EuroFIR European Network of Excellence and is included in the list of food composition databases of the EuroFIR Association (http://www.eurofir.net/eurofir_knowledge/european_databases). This information has been added to the methods section (Lines 169-182).

Although the amount of inorganic phosphate in beverages is specified in databases, but there is no precise information about the content of inorganic phosphate in solid foods. In processed food the phosphate content is largely inorganic thus it was accounted as inorganic. The estimation of inorganic phosphate intake was not as precise as the organic phosphate from animal or vegetable protein. Yet it was found a correlation between P/UUN and the content of inorganic phosphate in the diet. The issue is important and it is now included in the discussion of the article.

Round 2

Reviewer 3 Report

My main concern that I stated originally remains and it has not been answered by the authors. They have no precise way to estimate the inorganic P content from the food sources. Therefore, their conclusions in terms of the influence of inorganic P in determining urinary P are just theoretical ones and not based in facts as the conclusions suggest. I agree with their theoretical and logical analysis but in practical terms they have not demonstrated anything.

Author Response

REVIEWER 3

My main concern that I stated originally remains and it has not been answered by the authors. They have no precise way to estimate the inorganic P content from the food sources. Therefore, their conclusions in terms of the influence of inorganic P in determining urinary P are just theoretical ones and not based in facts as the conclusions suggest. I agree with their theoretical and logical analysis but in practical terms they have not demonstrated anything.

The reviewer is absolutely correct:

  1. we have no precise way to estimate the inorganic P content from the food sources” .

The new version of the manuscript includes a sentence (lines 192-196) stating that  “with the information available it is not possible to  estimate with precision the amount of inorganic phosphate contained in the diet; the data collected and the values reported are only a brief estimation based in the amount of processed food ingested (self-reported) by the patients.” .

2 “their conclusions in terms of the influence of inorganic P in determining urinary P are just theoretical ones and not based in facts as the conclusions suggest”.

We had available a self-reported questionnaire from patients indicating whether they have eaten processed foods, bakery, beverages and others that are known to contain different amounts of inorganic phosphate. Patients were asked to answer whether they ingested known processed food in a binary fashion (Yes/No). This is not an estimation of the amount of inorganic phosphate ingested; again, this is yes/no answer.  A logistic regression was performed to analyze if the urinary P/UUN ratio was associated with the probability of having consumed processed food. The mean (SD) values of P/UUN were: 71.4 ± 15.9 mg/g and 60.6 ± 14.9 mg/g, p=0.005, in patients that respectively reported having consumed processed food and those that did not.

Results of logistic multivariable regression analysis using “eat” vs “did not eat” processed food as dependent variable showed the following information:

Model 1

Model 2

Characteristic

OR

95 % CI

p-value

OR

95% CI

p-value

Gender (F vs. M)

0.939

0.232, 3.696

0.93

Age (Years)

0.885

0.811, 0.95

0.002

0.883

0.807, 0.95

0.003

P/UNN ratio (mg/g)

1.058

1.015, 1.111

0.013

1.061

1.019, 1.112

0.007

P from animal sources (mg/day)

1.000

0.997, 1.003

0.82

P from vegetal sources (mg/day)

1.002

0.997, 1007

0.52

GFR (CKD-EPI) (ml/min/1.73 m2)

1.000

0.97, 1.029

0.99

The P/UUN ratio is independently associated with the probability of having consumed phosphate from inorganic sources. Age, on the contrary, was associated with the probability of having consumed phosphate from other sources. The AUC of the model is 0.809 (shown below). Sensibility and specificity of the model are  82% and 64% respectively. The youden index with bootstrap estimation for simulating the cut point variability (1000 replications) was used to estimate the optimal cut point to discriminate patients that self-reported having consumed processed food from those who did not. Patients with a urine P/UUN ratio > 67.3 mg/g. have a 80% probability of eating  processed food.

Figure 3. ROC curve representing the diagnostic accuracy of model 2.

All this information has been added to the new version of the manuscript. (lines 261-269 and 342-374)

To obtain additional information about the relationship between inorganic phosphate intake and the ratio P/UUN in urine we have analyzed urine samples stored in our Biobank. These are samples from previous experiments using experimental rat model of 5/6 Nx fed diets with inorganic phosphate added to achieve a 0.2%, 0.4%, 0.6% and  0.8% of total dietary content of phosphate. The normal phosphate content in rat diet is 0.6%.

The results obtained are presented in the figure 4 showing that P/UNN in urine increases significantly with the percentage of inorganic P ingested.

These results support that P/UNN could be useful to estimate the intake of inorganic phosphate relative to other sources of phosphate.

The following text has been included in the new version of our manuscript:

Methods (Lines 225-253)

2.6.1 Animals

Male Wistar rats (Charles River Laboratories,Wilmington, MA, USA), 9-10 weeks old and weighing 250-300 g, were individually housed using a 12-hour/12-hour light/dark cycle and given ad libitum access to 1318 Altromin breeding diet (Altromin, Germany). Rats received humane care in compliance with the Principles of Laboratory Animal Care formulated by the National Society for Medical Research. Ethics approval was obtained from the Ethics Committee for Animal Research of the University of Cordoba.

 Renal failure was induced by subtotal nephrectomy (5/6 Nx), a two-step procedure. In the first step, animals were anesthetized using xylazine (5 mg/kg intraperitoneally) and ketamine (80 mg/kg intraperitoneally). A 5 to 8 mm incision was made on the left mediolateral surface of the abdomen. The left kidney was exposed, and the two poles (two thirds of the renal mass) were ablated. After 1 week, animals underwent right Nx under anesthesia. The control group underwent sham operation.

2.6.2 Dietary inorganic phosphorus modulation

One day after the second surgery, rats were randomly distributed into groups and fed diets with inorganic phosphate added to achieve a 0.2%, 0.4%, 0.6% and 0.8% of total dietary content of phosphate for 15 days. During the last 3 days of the 15-day experiment, rats were housed in metabolic cages to collect urine samples. Urine Urea nitrogen and phosphate were quantified by spectrophotometry (Biosystems, Barcelona, Spain).

Results (Lines 392-402)

To obtain additional information about the relationship between inorganic phosphate intake and the ratio P/UUN we used an experimental rat model of uremia by 5/6 nephrectomy fed diets with increasing amounts of inorganic P (0,2%, 0,4%, 0,6% and  0,8%) .Figure 4 shows that P/UNN increases significantly with the percentage of inorganic P ingested. This result supports that the ratio P/UUN could be useful to evaluate an excessive intake of inorganic phosphate.

Figure 4. Relationship between inorganic P intake and P/UNN ratio.  P/UUN was significantly higher as inorganic phosphate intake increases. a p<0.001 vs 0,2% group. b p<0.001 vs 0,4% group. c p<0,01 vs. 0.6% group. Values are  mean ± standard deviation.

Discussion (Lines 407-412 and 468-481)

Results obtained from animal studies demonstrate that adding inorganic phosphate to the diet produce a commensurate increase in the ratio P/UUN in urine.

The results obtained from groups of uremic rats fed increasing amount of inorganic phosphate indicate that maintaining phosphate intake of animal or vegetable origin the addition of known amounts of inorganic phosphate to the diet will produce predictable increases in  the P/UUN ratio in urine.

  1. I agree with their theoretical and logical analysis but in practical terms they have not demonstrated anything

We thanks the reviewer for agreeing with the theory behind our analysis and hope that these additional analysis presented in the new version of the manuscript demonstrate the utility of measuring the ratio P/UUN in urine, information that, to our knowledge, was not previously reported.
